# Repression of the Hox gene *abd-A* by ELAV-mediated Transcriptional Interference

**Javier J. Castro Alvarez**[1], **Maxime Revel**[1], **Judit Carrasco**[2,3], **Fabienne Cléard**[1], **Daniel Pauli**[1], **Valérie Hilgers**[2], **François Karch**[1], **Robert K. Maeda**[1]*

1 Department of Genetics and Evolution, University of Geneva, Geneva, Switzerland, 2 Max-Planck-Institute of Immunobiology and Epigenetics, Freiburg, Germany, 3 Faculty of Biology, Albert Ludwig University, Freiburg, Germany

* robert.maeda@unige.ch

## Abstract

Intergenic transcription is a common feature of eukaryotic genomes and performs important and diverse cellular functions. Here, we investigate the *iab-8* ncRNA from the *Drosophila* Bithorax Complex and show that this RNA is able to repress the transcription of genes located at its 3' end by a sequence-independent, transcriptional interference mechanism. Although this RNA is expressed in the early epidermis and CNS, we find that its repressive activity is limited to the CNS, where, in wild-type embryos, it acts on the Hox gene, *abd-A*, located immediately downstream of it. The CNS specificity is achieved through a 3' extension of the transcript, mediated by the neuronal-specific, RNA-binding protein, ELAV. Loss of ELAV activity eliminates the 3' extension and results in the ectopic activation of *abd-A*. Thus, a tissue-specific change in the length of a ncRNA is used to generate a precise pattern of gene expression in a higher eukaryote.

**Data Availability Statement:** All relevant data are within the manuscript and its Supporting Information files. Additional information can be provided if requested.

## Author summary

Although all of the cells making up complex organisms contain the same genetic material, they are nevertheless able to create the diverse tissues of the body. They do this by changing the genes they express. Thus, understanding how genes are controlled in a tissue-specific fashion is one of the primary interests of molecular genetics. Within the *bithorax* homeotic complex of the fruit fly *Drosophila melanogaster*, we, and others, previously showed that a >92 kb-long non-coding RNA, called the *iab-8* ncRNA, downregulates many important developmental genes, including its genomic downstream neighbor, the homeotic gene *abd-A*. This downregulation is important as its loss is linked to female sterility. Interestingly, we find that the *iab-8* ncRNA regulates *abd-A* through a mechanism called transcriptional interference, where one gene downregulates a target gene by transcribing over it. In the case of *iab-8*, this process is limited to the posterior central nervous system, where the *iab-8* ncRNA is specifically extended into the *abd-A* gene by the action of the neuronal-specific RNA binding protein, ELAV. Overall, our work highlights a largely unexplored mechanism by which tissue-specific gene regulation is achieved.

**Funding:** This work was supported by the Canton of Geneva (R.K.M and F.K.), the Swiss National Fund for Research (http://www.snf.ch/Seiten/VariationRoot.aspx) grant 31003A_149634 (to F.K. and R.K.M.) and grant 310030_192621 (to R.K. M)., the Claraz Foundation (to F.K., R.K.M. and D. P.), the Max Planck Society (https://www.mpg.de), the Deutsche Forschungsgemeinschaft (DFG, German Research Foundation (https://www.dfg.de) Project-ID 403222702 - SFB 1381 (to V.H.) and the European Research Council (ERC) under the European Union's Horizon 2020 research and innovation program ((https://erc.europa.eu) grant agreement no. ERC-2018-STG-803258 (to V.H.). Salaries for J.J.C.A., R.K.M, D.P. and F.K. were paid by the Canton of Geneva. The salaries of F.C. and J. J.C.A. were paid by the Swiss National Fund for Research (http://www.snf.ch/Seiten/VariationRoot. aspx) grant 31003A_149634 (to F.K. and R.K.M.). The salary of M.R. was paid by the Swiss National Fund for Research (http://www.snf.ch/Seiten/VariationRoot.aspx) grant 310030_192621 (to R.K. M). The salary of J.C. was paid by the European Research Council (ERC) under the European Union's Horizon 2020 research and innovation program ((https://erc.europa.eu) grant agreement no. ERC-2018-STG-803258 (to V.H.). And the salary of V.H was paid by the Max Planck Society (https://www.mpg.de). The funders had no role in the study design, data collection and analysis, decision to publish, or preparation of the manuscript.

**Competing interests:** The authors have declared that no competing interests exist.

## Introduction

Several noncoding RNAs (ncRNAs) have been identified from the Hox clusters of different species; a few of these have been shown to play key roles in gene regulation [1–10]. One of these ncRNAs is the 92 Kb, spliced and polyadenylated transcript called the *iab-8* ncRNA. Located within the *Drosophila* Bithorax Complex (BX-C), the *iab-8* ncRNA originates from a promoter located about 4.5Kb downstream of the *Abd-B* transcription unit and continues until within about 1 Kb of the *abd-A* promoter. *In situ* hybridization experiments show that it is transcribed specifically in the very posterior epidermis of the embryo from the cellular blastoderm stage. From later embryonic stages, its expression becomes limited to parasegments (PS) 13 and 14 of the CNS [1, 2, 11–14]. Loss of the *iab-8* ncRNA has been shown to result in both male and female sterility, likely due to problems in the innervation of muscles important for reproduction [14, 15]. Much of its function has been attributed to a microRNA located between its sixth and seventh exons, called *miR-iab-8* (miRNA). *miR-iab-8* targets multiple transcripts including the *Ubx* and *abd-A* homeotic genes and their cofactors *hth* and *exd* [1, 15–17]. Indeed female sterility has been directly linked to ectopic *hth*, *Ubx* and *abd-A* in the CNS[1, 15–17].

In the embryonic CNS, *abd-A* expression is normally limited to PS7-12. Studies performed by our lab and others have shown that the restriction of *abd-A* expression from PS13 in the CNS is dependent upon expression of the *iab-8* ncRNA [1, 14, 15, 18]. Although, the *miR-iab-8* miRNA plays a part in the repression of *abd-A* in PS13, a deletion of the miRNA template sequence only results in a mild derepression of *abd-A* in PS13 (see **Fig 1D**) [14]. On the contrary, mutations preventing the production of the *iab-8* ncRNA cause a complete de-repression of *abd-A*, such that the *abd-A* expression pattern in PS13 mimics that of PS12 (see **Fig 1C**), suggesting the existence of a second repression mechanism. Here, we explore the mechanism by which the *iab-8* ncRNA represses *abd-A*. Using deletions spanning different regions of the *iab-8* transcript, we were unable to identify specific parts of the transcript that can account for the additional repression of *abd-A* by the *iab-8* ncRNA. Furthermore, we find that the *iab-8* transcript can repress an exogenous reporter gene placed downstream of its sequence. Based on these findings, we conclude that it is the act of transcription that is necessary for repression, rather than the sequence transcribed (a phenomenon called transcriptional interference). Examination of the *iab-8* transcript in the CNS, shows that there is a 3' extension made specifically within the CNS. This elongated transcript seems to be essential for *abd-A* down-regulation and requires the neuronal-specific, RNA-binding protein ELAV (or its paralogue FNE) for its creation. Overall, our work suggests that ELAV mediates a 3' extension of the *iab-8* ncRNA that, in turn, allows it to specifically repress *abd-A* expression in the posterior CNS through transcriptional interference.

## Results

### CRISPR-mediated deletions within the transcriptional unit of the *iab-8* ncRNA do not affect the microRNA-independent repression of *abd-A* in the CNS

In order to identify the second element in the *iab-8* transcript that is required for *abd-A* repression, a series of CRISPR-mediated deletions were created within the *iab-8* ncRNA template. Previously, we used classically isolated deletions and chromosomal breaks to eliminate much of the 5' sequences as containing this second activity. From these studies, we were able to eliminate all but the final two exons of the *iab-8* ncRNA [14]. Here, we examined deletions

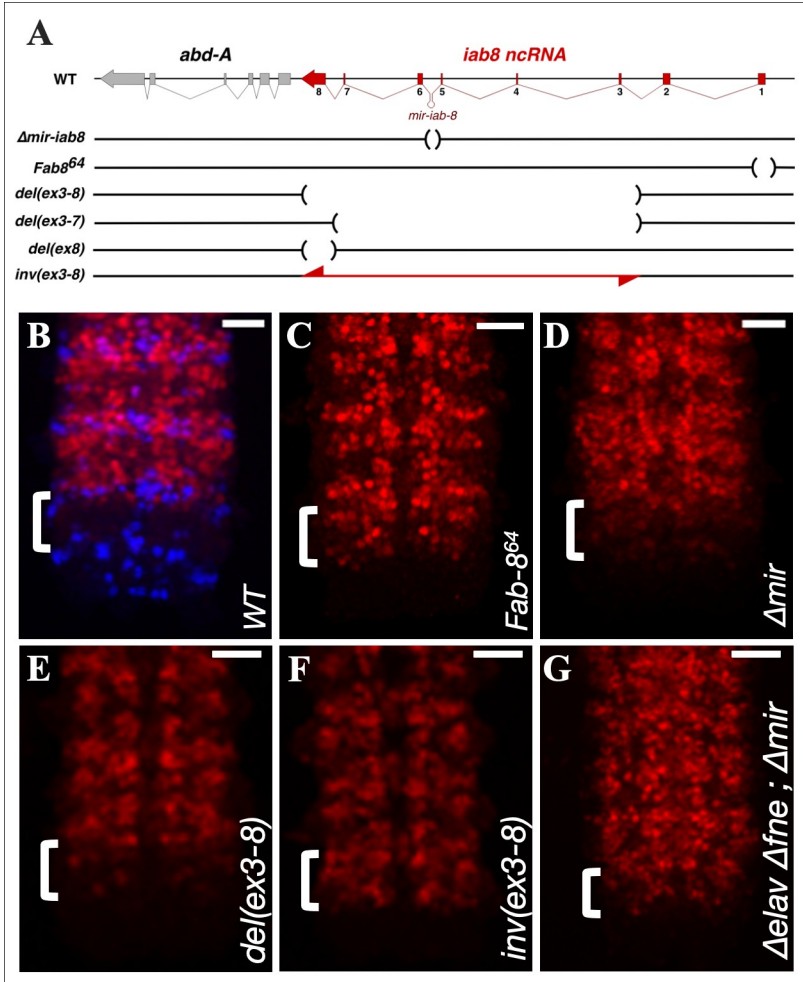

**Fig 1. Deletions and inversions within the *iab-8* ncRNA affect ABD-A expression in PS13. A.** Shows a schematic representation (not to scale) of the mutants used in this study. Regions deleted are marked by parentheses in the horizontal line and the regions inverted are shown by a double-arrowed red line. **B.-G.** Shows Z-projection images made from confocal stacks of the posterior, stage 13/14 CNS stained with anti-ABD-A (red) and EN (blue in **B.** to show parasegments) Genotypes of the embryos are indicated at the bottom right corner of each panel. Scale bar = 20 μm.

that remove large portions of the *iab-8* ncRNA starting from the 3' end, but leave the *iab-8* promoter and first two exons intact for continual transcription. The largest of the deletions examined was *del(ex3-8)* [19]. This deletion leaves only the first two exons of the *iab-8* ncRNA, leading to a juxtaposition of these exons with the *abd-A* promoter (**Fig 1A**). Although this deletion removes almost the entirety of the *iab-8* transcription unit, including the *iab-8* miRNA, *del(ex3-8)* shows only a mild de-repression of *abd-A* in PS13 of the CNS (**Fig 1E**). By contrast, a deletion that removes the *iab-8* ncRNA promoter (*Fab-8^{64}*) shows a complete de-repression of *abd-A*, such that *abd-A* expression in PS13 resembles that found in PS12 (**Fig 1C**). In fact, the mild de-repression seen in *del(ex3-8)* is reminiscent of the phenotype caused by the clean deletion of *mir-iab-8* alone (**Fig 1D**), which is also deleted in this mutant. Since the *del(ex3-8)* mutation does not show more extensive *abd-A* derepression than the deletion of *mir-iab-8* alone, we conclude that the sequence deleted in *del(ex3-8)* does not contain the secondary repressive element encoded by the *iab-8* ncRNA.

Based on the analysis of *del(ex3-8)*, the second repressive element should be located upstream of exon 3 in the *iab-8* transcript. This was at odds with our previous results, where we concluded that the 3' region of the transcript was important for *abd-A* repression. Analysis of a chromosomal inversion of the *iab-8* exon 3–8 region, helped to shed light on this discrepancy. During the course of generating our deletions, we also obtained, at high frequency, inversions of the areas targeted for deletion (**Fig 1A**). One of these deletions, inverts the exact sequences deleted in *del(ex3-8)* that we call *inv(ex3-8)* [19]. In *inv(ex3-8)*, the *iab-8* promoter and first two exons are untouched and, by *in situ* hybridization using probes directed against these exons, seem to be transcribed normally (**S1 Fig**). We reasoned that if these exons produce something that is important for *abd-A* repression, *inv(ex3-8)* should repress *abd-A* about as well as *del(ex3-8)*. However, examining *abd-A* expression in *inv(ex3-8)*, we see a total derepression of *abd-A* in PS13 of the CNS (**Fig 1F**), thus suggesting that the sequences of exons one and two are not, by themselves, crutial for repressing *abd-A*.

Based on the data from the deletions and inversions, no particular sequence can account for the miRNA independent repression of *abd-A*. We cannot, however, rule out the importance of both the *iab-8* and *abd-A* promoter regions. Previously, we published 3'RACE results that showed that about 5% of the *iab-8* transcripts in embryos do not terminate at the end of exon 8 [14]. We wondered if this "read-through" transcription could be important for *abd-A* repression. To visualize this transcription, we performed *in situ* hybridization using a probe against the intergenic sequence located between the terminal exon of the *iab-8* ncRNA and *abd-A*. As seen in **S2 Fig**, in wild type embryos, significant transcription of this region can be detected in PS13 and 14 of the CNS (**S2B Fig**). Using this same probe on *del(ex3-8)* and *inv (ex3-8)* embryos, shows that transcription of this area is still present in *del(ex3-8)* embryos (**S2C Fig**), but is lost in *inv(ex3-8)* embryos (**S2D Fig**). It is interesting to note that in *wild-type*, although the *iab-8* ncRNA can be detected using an exon 8 probe in the epidermis, the intergenic transcription, can only be seen in the developing CNS, which probably accounts for the low proportion of this transcript detected in the 3'RACE studies. Thus, these results suggest that there is a tissue-specific extension of the *iab-8* ncRNA that allows the *iab-8* transcript to extend past exon 8 towards the *abd-A* promoter region. Moreover, this extended transcript is eliminated in the *inv(ex3-8)* embryos, where *abd-A* repression is lost.

## CNS-specific transcriptional read-through of the *iab-8* ncRNA

The probe directed against the intergenic region between the *iab-8* ncRNA and *abd-A*, demonstrates that there is significant transcriptional read-through from the *iab-8* ncRNA in PS13 and 14 of the embryonic CNS. Examining recently published RNA-seq data on timed embryo collections, the presence of the readthrough transcripts can be verified (**see below**) [20]. In order to explore the extent of the transcriptional read-through, we tested additional probes within the *abd-A* transcription unit. The most-3' probe was generated against the 3'UTR of *abd-A*. This probe detects *abd-A* transcription in the CNS, in a domain that spans from PS7 to PS14 (**Fig 2**). This domain of transcription extends beyond the normal PS7-PS12 domain of expression found by ABD-A protein immunostaining. Interestingly, the transcript signal detected with the *abd-A* 3'UTR probe shows two distinct patterns: in PS7 to PS12, the domain that corresponds to ABD-A protein expression, the staining is concentrated in certain clusters of cells, giving a characteristic, aspect to each parasegment that we call the "domino" pattern (**Fig 2B, black arrows**). Meanwhile, in PS13 and PS14, a diffuse, uniform signal is observed (**Fig 2B, grey arrows**).

No previous evidence has shown that the *abd-A* promoter is highly active in PS13-PS14 of *WT* embryos. In light of our observation of high levels of read-through transcription in the

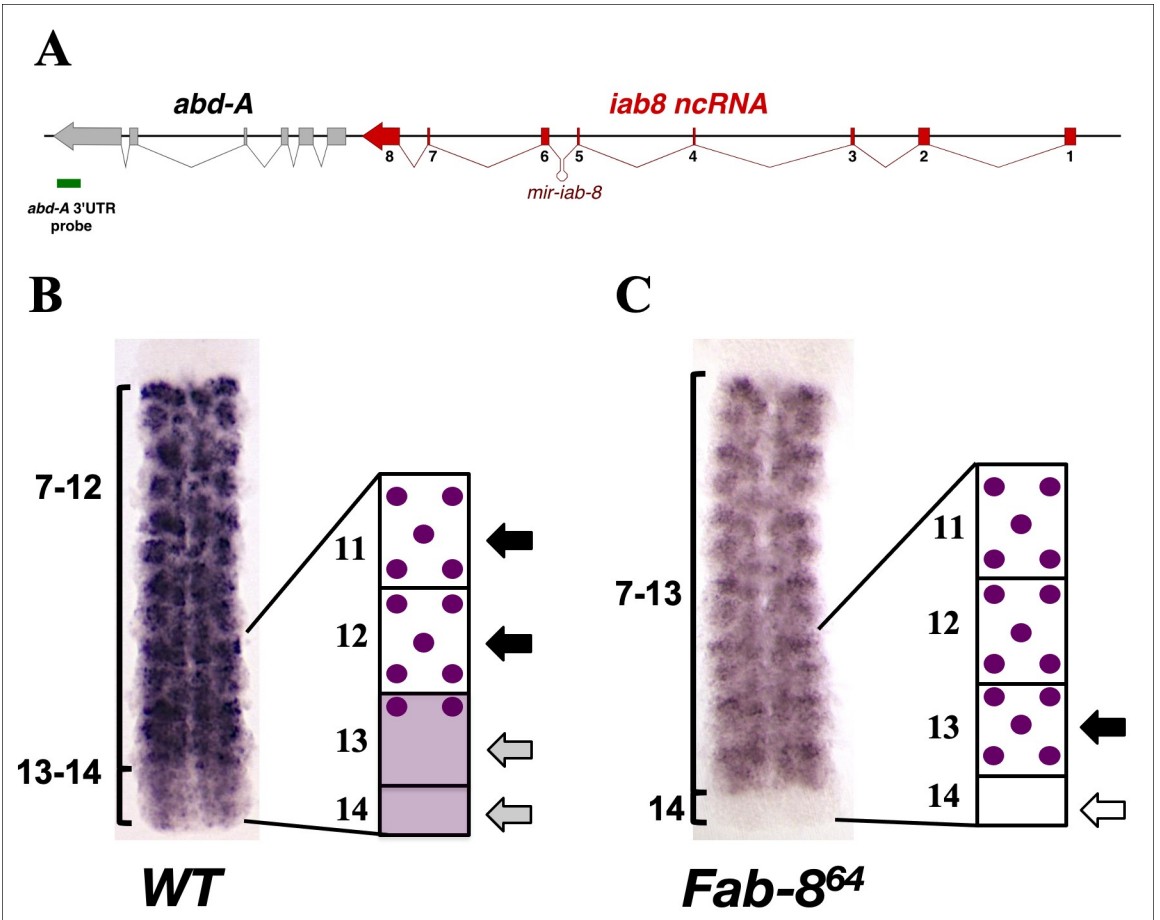

**Fig 2. In situ hybridization using an *ABD-A 3'UTR probe* indicates read-through transcription from the *iab-8* ncRNA in PS 13 and 14. A.** Shows a schematic representation (not to scale) of the genomic region with the location of the probe marked by a green bar beneath the genes. The stage 13/14 CNS from a *wild-type* (**B.**) or *Fab-8^64* (**C.**) embryo hybridized with the *ABD-A 3'UTR* probe and a schematic representation of this staining. Parasegment areas are labeled. The "domino" pattern is schematized in the drawings to the right of the stained tissues and is indicated by the black arrows. The diffuse pattern coming from the *iab-8* promoter is seen in PS 13 and 14 and schematized as a shaded area in the diagram next to the *wt* CNS and indicated by the grey arrows. The loss of this diffuse pattern is seen in the mutant CNS and is indicated in the diagram next to the mutant CNS by a white arrow.

CNS and the different staining pattern in the most-posterior parasegments, we asked if the staining observed in these parasegments could correspond to transcriptional activity emanating from the different promoters (*abd-A* vs *iab-8*). To test this, we performed *in situ* hybridizations using the same *abd-A* 3'UTR probe on *Fab-8^64* embryos, which lack the *iab-8* ncRNA promoter. In these mutants, we observe that the uniform signal normally present in PS13 and 14 (**Fig 2C**, **white arrow**) is gone. However, the "domino" pattern remains and actually invades PS13 (**Fig 2C**, **black arrow**). These results are consistent with the idea that the diffuse pattern represents transcriptional read-through by the *iab-8* ncRNA into the *abd-A* transcription unit and that, in the absence of the *iab-8* ncRNA, the *abd-A* enhancers are able to drive expression of *abd-A* in PS13 (but not PS14).

In order to show that the *iab-8* ncRNA continues into the *abd-A* transcription unit within the CNS, we performed RT-PCR experiments on 0–24 hour-old embryos to identify *iab-8/abd-A* fusion transcripts. The RT-PCR experiments were performed using primers in different exons of the *iab-8* ncRNA and *abd-A*. Using this method, five differentially spliced isoforms

were identified; all of the isoforms were composed of 5' exons from the *iab-8* ncRNA and 3'exons of *abd-A*. For the most part, the alternative isoforms were generated by splicing from exons 6 or 7 of the *iab-8* ncRNA, to exons 2, 4 or 5 of *abd-A*. All variants lacked exon 8 of the *iab-8* ncRNA and all but one isoform lacked the initial ATG of *abd-A* (*iab-8(7)-abd-A(2)*) (see **Fig 3,** and also the discussion regarding the ATG-containing isoform).

## Transcription of the *iab-8* ncRNA represses the expression of a downstream reporter gene

Thus far, our results indicate that the *iab-8* ncRNA extends well into the *abd-A* gene in PS13 and 14 of the CNS. Furthermore, we see read-through transcription in embryos where *abd-A* is repressed (like wild type and *del(ex3-8)* embryos), but not in cases where *abd-A* is ectopically expressed (like *Fab-8*[64] and *inv(ex3-8)*). If *abd-A* is repressed by the extension of the *iab-8* transcript, then one likely mechanism by which this could occur is transcriptional interference. Transcriptional interference relies on the act of transcription across a target promoter/gene to repress its transcription, presumably by either physically removing a downstream RNA polymerase or changing the chromatin environment to make it less capable of initiating transcription. Thus, it is independent of the sequence of both the repressing and the targeted transcripts. To determine if transcriptional interference could explain the additional repression of *abd-A* in the embryonic CNS, we decided to recreate the transcriptional situation present at the *abd-A* promoter, using a reporter system that would be independent of trans-acting factors produced by the ncRNA.

In the reporter line, we fused the *EGFP* cDNA to the *abd-A* promoter and the first 140bp of the *abd-A* 5'UTR. We then integrated this construct (which also contains *iab-8* exon 8) into the genomic locus of exon 8 of the *iab-8* ncRNA (*abd-A:GFP*: **Fig 4**) to recreate the *abd-A* promoter region downstream from the endogenous *iab-8* ncRNA in the BX-C, but with the *EGFP* sequence replacing *abd-A*. This modified gene should respond to the BX-C enhancers that drive localized expression of *abd-A*, but, since it does not contain the vast majority of normal *abd-A* sequence (including its 3'UTR), should not be repressed by any sequence-specific, *trans*-acting factors produced by the ncRNA (like *mir-iab-8*).

Examining EGFP expression in these embryos, we see that the *EGFP* reporter, like the wild type *abd-A* protein (**Fig 4B**), is strongly expressed in the embryonic CNS from PS7 to PS12

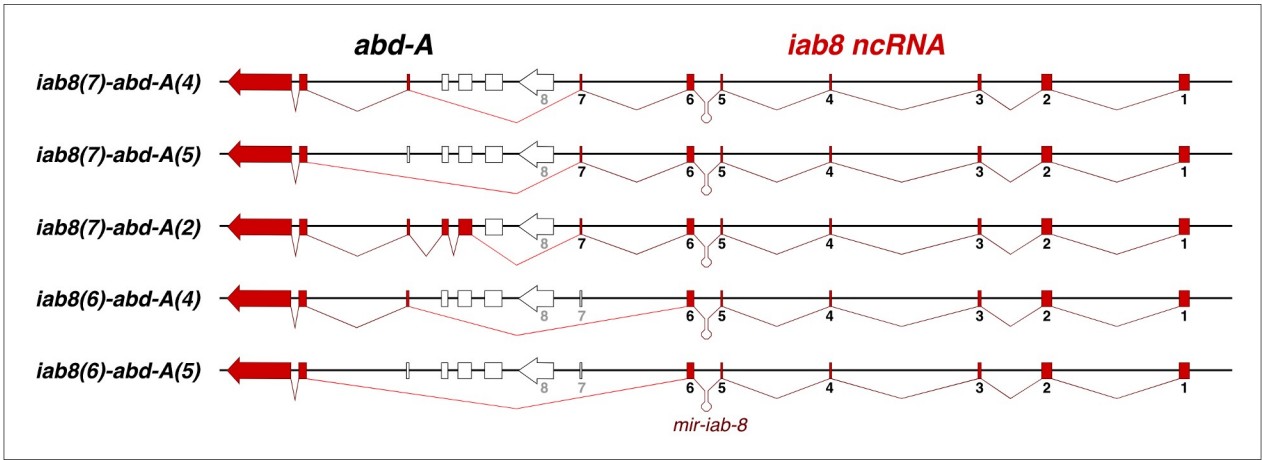

**Fig 3. Splice variant found between the *iab-8* ncRNA and *abd-A* transcription units in the CNS.** A schematic representation of the genomic region is shown with the different exons represented by broken block arrows. The utilized exons of the *iab-8 ncRNA* are shown in red and the utilized exons of *abd-A* are shown in pink. Unused exons are white. The different splice products are shown by the red lines between exons.

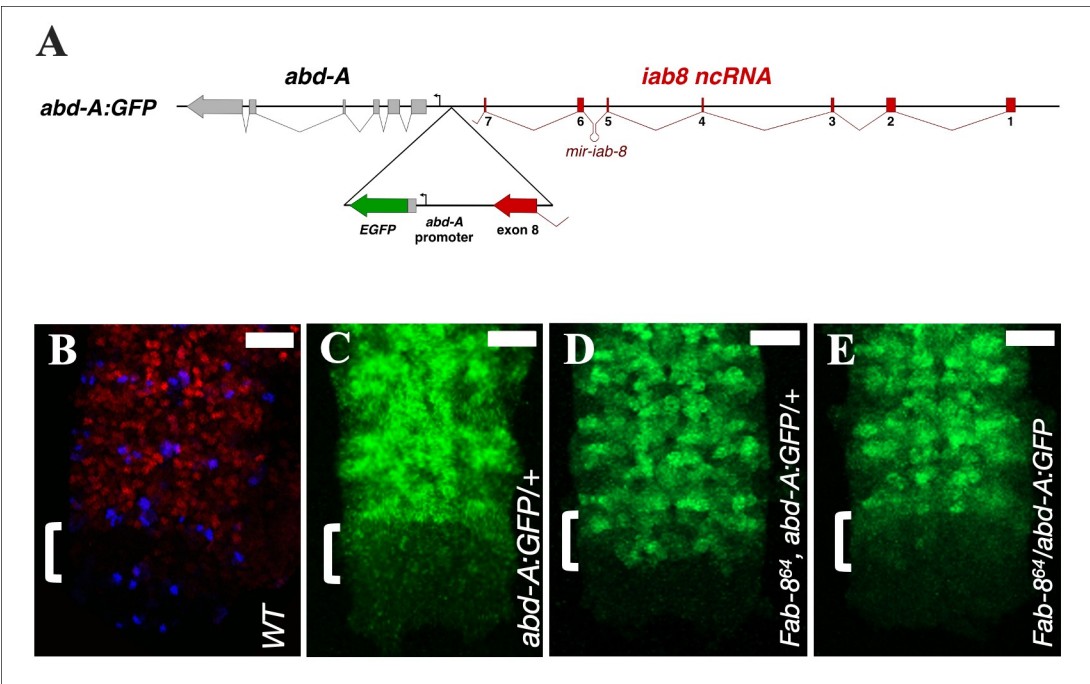

**Fig 4. *abd-A*:*GFP* expression in different genetic contexts. A.** A magnification of the *abd-A*-EGFP reporter construct inserted downstream from exon 8 is displayed in the triangle below the broader genomic context. The resulting construct consists of a duplication of the exon-8-*abd-A* interval. The EGFP coding sequence starts after 140 bps of the *abd-A* 5'UTR. **B.-E.** Each panel shows a Z-projection image made from confocal stacks of the posterior stage 13/14 CNS stained with: anti-ABD-A (in red), anti-EN (in blue to show the anterior parasegment boundaries) and anti-GFP (in green). Genotypes of the embryos are indicated in each panel. Scale bar = 20 μm.

(**Fig 4C**). However, unlike the wild type ABD-A protein, there is a slight ectopic expression of EGFP in PS13 of the CNS (**Fig 4C, bracket**). This ectopic expression is similar to the pattern of ABD-A in *miR-iab-8* mutants (**Fig 1D**). As the *abd-A*:*GFP* reporter lacks microRNA binding sites, this result was expected. Importantly, however, EGFP expression in PS13 is significantly lower than that seen in PS12. Thus, the second repressive mechanism is still present and can prevent the full expression of GFP in this parasegment. To prove that this repressive mechanism is dependent on *iab-8* transcription, we then recombined an *iab-8* promoter deletion (*Fab-8⁶⁴*) onto the *abd-A*:*GFP* chromosome. Strikingly, this leads to a full de-repression of EGFP in PS13 of the CNS (**Fig 4D**), proving that the expression of the *iab-8* ncRNA is required for the repression of the *abd-A*:*GFP* gene in this parasegment. Finally, as transcriptional interference requires the interfering transcript to be on the same chromosome, we compared the expression of the GFP reporter when it was placed in *cis* or in *trans* to the *iab-8* promoter deletion. As seen by comparing **Fig 4D** to **Fig 4E**, the complete derepression of the GFP reporter requires that the deletion be placed in *cis* to the reporter. Based on these experiments, we conclude that transcriptional interference limits the spatial domain of *abd-A* expression.

### The neuronal specific gene ELAV enhances the production of the transcriptional read-through of the *iab-8* ncRNA and is necessary for transcriptional interference of *abd-A*

As mentioned above, the read-through transcription of the *iab-8* ncRNA seems to be tissue-specific and is restricted to the CNS. Indeed, while a probe against exon 8 of the *iab-8* ncRNA

is able to detect expression of this gene in the ectoderm of early embryos (**Fig 5B**) and the CNS of late embryos (**Fig 5C**), none of the other probes targeting the genomic areas downstream from exon 8 of the *iab-8* ncRNA is able to detect significant transcription in PS13-14, in any tissues outside of the CNS (**Fig 5**).

The CNS-specific nature of this transcriptional read-through made us wonder if it could be due to the action of a neuronal tissue-specific factor. The neuronal RNA binding protein ELAV has previously been described [21] as a protein responsible for the alternative polyadenylation and splicing of neuronal-specific isoforms of numerous genes. In fact, the homeotic genes, including *abd-A*, are known targets of ELAV, whose activity causes an elongation of the 3'UTR, often making them more susceptible to miRNA regulation [22]. Following this idea, we examined the expression of the *iab-8* ncRNA in *elav* mutants by *in situ* hybridization using probes targeting two different areas of the main transcription unit (*iab-8ex1-2* and *iab-8ex8*) and an additional probe targeting the intergenic region between *iab-8* and *abd-A*. As illustrated in **Fig 6**, although *elav* mutants show similar levels of transcription emanating from the *iab-8* ncRNA promoter (as visualized by the *iab-8ex1-2* and *iab-8ex8* probes, **Fig 6A–6D**), they show a strong reduction in the amount of transcript that continues past *iab-8* exon 8 (as

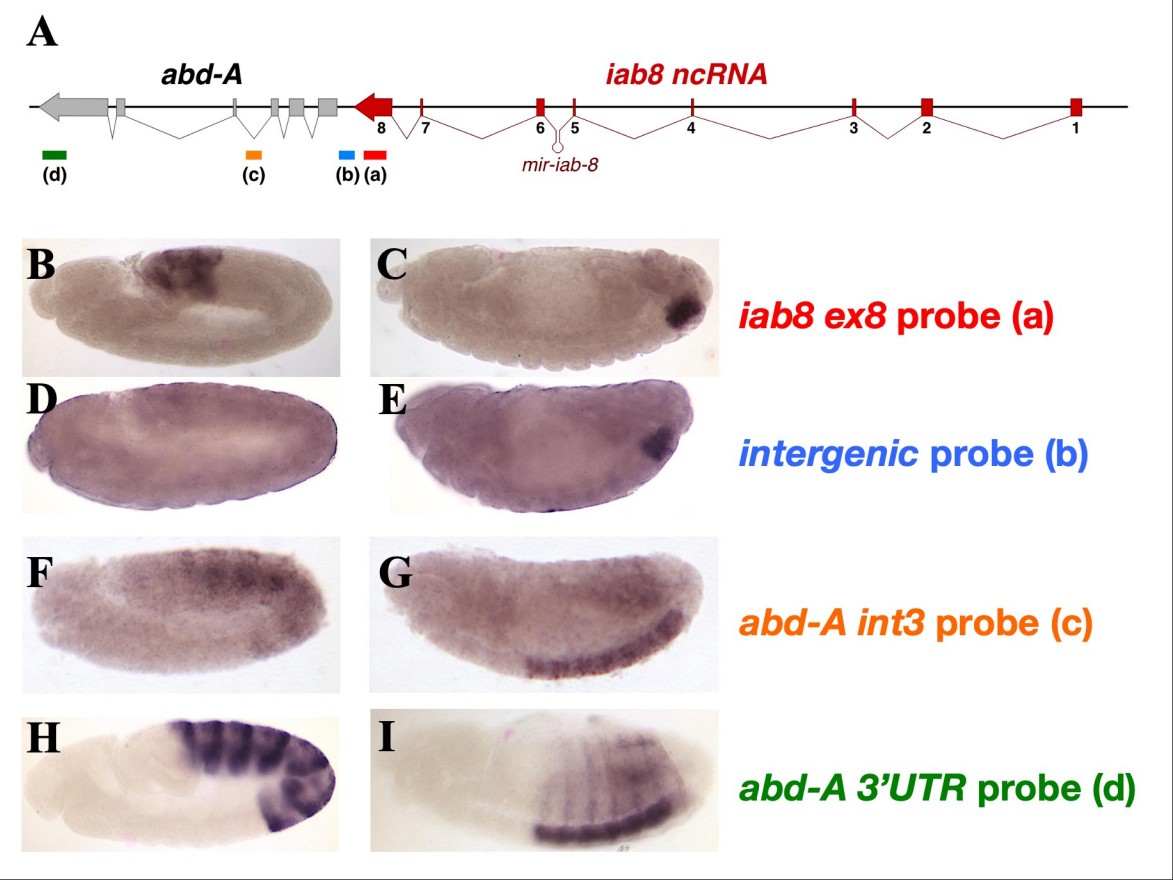

**Fig 5. Detection of the *iab*-8-ncRNA expression in wild-type embryos by different probes.** Wild type embryos were hybridized with probes directed against different area of the *iab-8/abd-A* region. The locations of the probes are indicated by the colored lines (and letters a, b, c and d) beneath the genomic map. The names of the probes are indicated to the right of the embryos displayed in panels B-I in their respective colors. Panels **B, D, F** and **H** show germband extended embryos (about stage 9) while panels **C, E, G** and **I** show germband retracted embryos (stage 13–14).

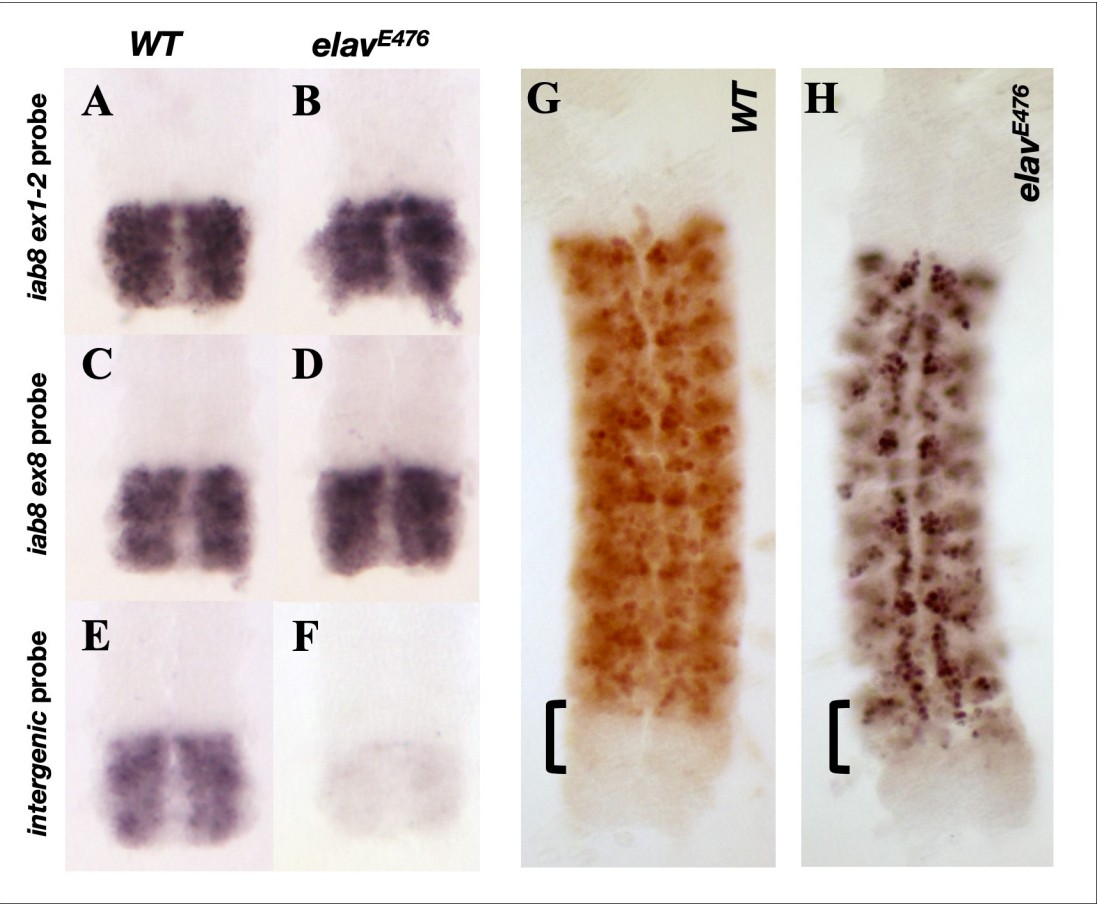

**Fig 6. Expression of the iab-8 ncRNA and ABD-A in dissected stage 13/14 CNSs in *wild-type* and *elav* mutants.** Panels **A.** through **F.** show dissected posterior nerve chords from *wild-type* and *elav* mutant embryos. Note that WT and mutant sample pairs were processed in parallel with their respective probes (the same as the probes used in Fig 5). Panels **G.** and **H.** display whole stage 13/14 dissected CNS immunostained for ABD-A (WT in **G** and *elav*^E476 in **H).**

visualized by staining with the *intergenic* probe, **Fig 6E and 6F**). Thus, ELAV is required, at least in part, to extend the *iab-8* transcript beyond exon 8.

Since *abd-A* repression in PS13 seems to depend on read-through transcription from the *iab-8* transcript and the read-through transcript seems to be ELAV dependent, it follows that *abd-A* repression should be ELAV dependent. We tested this by examining ABD-A expression in *elav* embryos. Indeed, immunostaining of *elav* embryos against ABD-A shows a noticeable level of ABD-A protein derepression in PS13 of the CNS (**Fig 6H**).

It is noteworthy that the *elav* embryos used in the experiments above still express the *mir-iab-8* miRNA. Also, recent studies have shown that in the absence of *elav*, its paralogue *fne* is differentially spliced to make a form of FNE capable of performing some of ELAV's functions [20, 23]. Our *in situ* results show that a small amount of read through transcription continues in *elav* mutants (**Fig 6F**). However, the recent transcriptomic analysis of *elav* mutants shows that there is a reduced, but noticeable amount of the read-through transcription in *elav* mutants. This amount of readthrough transcription is further reduced by also removing the *elav* homologue *fne*. This can be seen in Fig 7A and 7B, where RNA isolated from embryos of different ages were examined for read through transcription. To verify an effect on ABD-A expression, we examined ABD-A protein levels in embryos lacking *elav*, *fne* and *miR-iab-8*. As

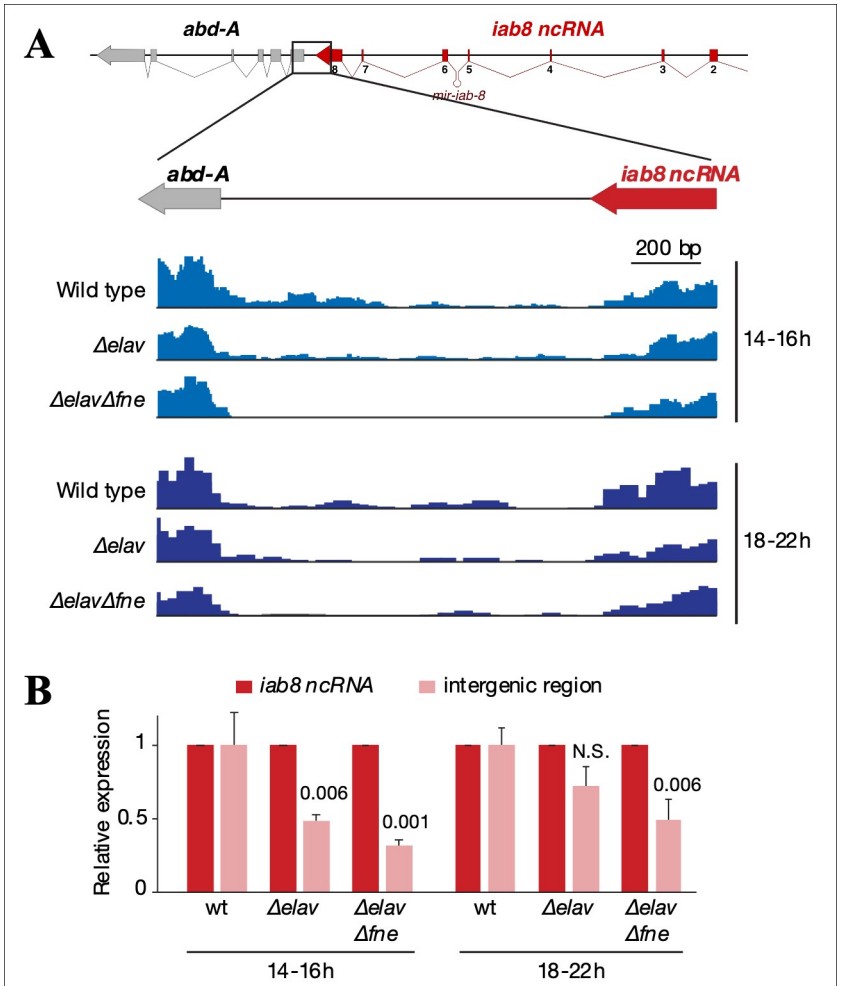

**Fig 7. The intergenic transcribed region is downregulated in *Δelav* and *ΔelavΔfne* embryos. A.** mRNA-seq signal tracks for the boxed gene region, representing expression of polyadenylated RNAs in whole embryos of the indicated genotypes, at two different times after egg laying (AEL). mRNA-seq data are from Carrasco et al., 2020 [20]. **B.** RT-qPCR quantification of *iab8* ncRNA and the intergenic region RNA in wild-type, *Δelav*, and *ΔelavΔfne* embryos. RNA was extracted from whole embryos 14-16h AEL (after egg laying) and 18-22h AEL. RNA levels were normalized to *iab8* ncRNA levels. Error bars represent mean ± SD of three biological replicates (10–15 embryos per replicate) for each genotype. P-values are indicated (one-way ANOVA testing the difference to wild type). As typical for ELAV-dependent RNAs, the intergenic region is more significantly downregulated in *ΔelavΔfne* than in *Δelav* embryos at both time points. N.S.: non-significant.

seen in **Fig 1G**, removal of these three elements is able to very strongly derepress *abd-A* in PS13 (see Discussion). Overall, these observations led us to conclude that the spatial restriction of *abd-A* expression in the CNS is determined by the *iab-8* miRNA along with transcriptional interference of the *abd-A* gene by the *iab-8* ncRNA in an ELAV/FNE dependent fashion.

## Discussion

Previously, we reported that the main transcript of the *iab-8* ncRNA terminates ~1 kb upstream of the *abd-A* transcription unit. This finding was based on 3'RACE experiments performed on RNA isolated from relatively early embryos (6–12 hours). Here we show that, in later embryos, where the *iab-8* ncRNA is restricted to the CNS, the *iab-8* transcript extends well into the *abd-A* transcription unit. In every condition tested, we see that *abd-A* is repressed when there is an

extended transcript. However, outside of the sequences required for normal transcription and RNA processing, the *iab-8* transcript, itself, does not seem to require any specific sequences to mediate this repression. Based on these findings, and the *cis* nature of this repression, we conclude that the act of transcribing the extended *iab-8* ncRNA is what represses *abd-A* expression in PS13 of the CNS. This type of inhibition is called transcriptional interference.

In transcriptional interference the transcription of one gene spreading over the coding or regulatory sequences of another gene is able to downregulate the target gene's expression [24]. The mechanisms mediating transcriptional interference seem to depend on the relative position of both promoters. In the case of the *iab-8* ncRNA and the repression of *abd-A*, we have a case of tandem promoters, where the genes are transcribed in the same direction and the upstream transcript transcribes over portions of the downstream gene (promoter, enhancers, transcription unit). Studies performed in single cell organisms (yeast and bacteria) suggest that there are two main mechanisms mediating transcriptional interference of tandem promoters. The first is called the "sitting duck" mechanism, where an initiating RNA polymerase or activating transcription factors are knocked off of the target gene by the passing polymerase. The potential second mechanism to mediate transcriptional interference is called the "occlusion" mechanism, where activating transcription factors (or RNA polymerase itself) for the downstream gene are prevented from binding to their binding sites by the passing RNA polymerase or by the modified chromatin structure following the passage of an elongating polymerase. Thus far, we cannot distinguish between these two mechanisms in our system. However, both mechanisms have been shown to be dependent upon the strength of the silencing transcript's promoter relative to the target transcript promoter. The stronger the promoter activity from the upstream gene, the stronger the repression of the downstream gene [25]. In the case of transcriptional interference by the *iab-8* ncRNA, we believe that its level of transcription is approximately equivalent to that of *abd-A*. Indeed, using an *abd-A* intronic probe to compare levels (a probe not subject to possible stabilization of the exonic probes of the *abd-A* coding mRNA), we see a similar level of transcription from both the *iab-8* (PS13 and 14) and *abd-A* promoters (PS7 through PS12) (**S3 Fig**). Given the slower nature of transcription initiation vs transcriptional elongation, this high level of transcription might favor downstream gene repression.

From work on mammalian cells, it has long been known that the final exon of coding genes often promotes termination by the recruitment of the termination machinery to the poly(A) site [26]. Although in recent years, ELAV has been studied as a protein whose function lies in extending the 3'UTRs of neuron-specific genes [21] by altering the selection of poly-A signals[27, 28], RT-PCR results suggest that, here, ELAV may play a role in the alternative splicing of the final exons of the *iab-8* transcript. This function in alternative splicing is consistent with the role described for ELAV as an RNA binding protein involved in the alternative splicing of the neuronal isoforms of the *Nrg* [29, 30] and *fne* gene products [20, 23]. In fact, ELAV family members have been shown to be particularly important for splicing into a terminal exon [31]. Thus, ELAV might extend the *iab-8* ncRNA by suppressing the ability of the *iab-8* transcript to splice into its normal terminal exon. This would then prevent the transcribing RNA polymerase from terminating, causing it to continue transcribing until it finds a new terminal exon. Published ChIP-seq experiments (where nascent transcripts were cross-linked to the genomic DNA along with proteins) on ELAV from early and later embryos support this interpretation. According to these results, there is additional ELAV binding at the junction between intron 7 and exon 8 of the *iab-8* transcript in later embryos when *iab-8* is expressed only in the CNS [32].

Interestingly, among the spliced fusions between *iab-8* and *abd-A*, we found one isoform that contains the *abd-A* ATG sequence. This would seem counter-productive, if the function of transcriptional interference is to prevent *abd-A* expression. Although we cannot judge the

amount of this specific transcript based on our experiments, previous results from our lab have suggested that exons one and two of the *iab-8* ncRNA act as translational repressors. Indeed, the MSA RNA [19], which is identical to the *iab-8* ncRNA except that the first two exons of *iab-8* are replaced by an alternative first exon, actually codes for a peptide whose coding sequence lies in the shared last exon [33]. A GFP fusion to this peptide and other reporters placed in the *iab-8* sequence have shown that these proteins are never expressed in the CNS, but can be expressed in the male accessory gland, where MSA is expressed[33]. Thus, even if this form is produced in a significant quantity, it seems that the embryos have further buffered themselves against ectopic *abd-A*, by repressing its translation.

Lastly, it is of note that even in the *elav*, *fne*, *mir-iab-8* triple mutant, the derepression of *abd-A*, while strong, may not be complete. There are a few cells, that still seem to repress *abd-A* in the posterior CNS (compare **Fig 1C** or **1F** with **1G**). At the moment, we cannot explain this result. We believe that some of this change may be due to fact that *elav*, *fne*, *miR-iab-8* mutant nerve cords are very much abnormal and may have certain cellular defects. We noticed, for example that these nerve cords were more difficult to dissect as they were to the extremely fragile relative to wild type. However, it is also possible that there are additional factors that allow transcriptional readthrough in these embryos or perform a repressive function on *abd-A* by another mechanism. Interestingly, our RT-qPCR results still seem to detect a low level of transcriptional readthrough even in *elav*, *fne* double mutants, hinting that some transcriptional interference might occur even in the absence of these factor. One possible candidate to mediate this transcriptional readthrough is the *rbp9* gene, the third *elav* homologue in *Drosophila*. Like *elav* and *fne*, *rbp9* that is expressed in neurons and has been shown to be capable of promoting 3'RNA extensions when ectopically expressed in cultured cells[23].

## Transcriptional interference and the Hox clusters

As a mechanism of transcriptional repression, transcriptional interference has mostly been found in organisms with compact genomes like yeast and bacteria. Because most of the multi-cellular eukaryotes studied in the lab have much larger genomes, containing a large proportion of "non-essential" DNA, transcriptional interference has often been disregarded as a common mechanism for gene repression. However, due to co-regulation and/or gene duplication events, eukaryotic genes may be more compact at certain locations than generally assumed. This is very evident in the HOX gene clusters where there are numerous examples of tightly packed or overlapping transcription units. With all of these examples of overlapping transcription units and possible transcriptional interference, it is interesting to ask if this association could relate to an ancient gene regulatory mechanism. Within the Hox genes there is a known phenomenon called posterior dominance. According to the principle of posterior dominance, the more posterior Hox gene expressed in a segment generally plays the dominant role in patterning the segment. In *Drosophila*, this is often seen by down-regulation of the more-anterior gene. It is interesting to note that in the most studied Hox clusters, the Hox genes are organized on the chromosome in a way in which each Hox gene is located directly 3' to the next more-posterior segment specifying Hox gene. If we consider that the Hox clusters are thought to have arisen from successive gene duplication events and after such duplication events, the two genes should have equal regulatory potential, then how could the upstream gene consistently take on a more dominant role? Transcriptional interference provides a possible explanation for this. According to this model, the upstream gene might have a slight advantage over the downstream gene due to transcriptional interference. This advantage, although potentially weak in many cases, could then be intensified and fixed through evolving cross-regulatory

interactions. In our case, the finding that ectopic *abd-A* in the posterior CNS leads to female sterility would help to drive such interactions.

Although we have studied this phenomenon in a HOX cluster, other situations might exist where genes are located in similar tight configurations that induce transcriptional interference. An interesting bioinformatic analysis of nested genes in *Drosophila* suggests that transcriptional interference might be a natural consequence of tight, tandem gene arrangement. In this study, the authors showed that there is a significantly lower number of nested genes transcribed from the same strand in the *Drosophila* genome[34]. Furthermore, nested gene in the same orientation contained fewer or no introns. Examining the expression data of the tandem, nested genes showed that these genes were often downregulated in tissues where the upstream gene was expressed, leading the author to suggest that the genetic arrangement of the genes might lead to transcriptional interference through mechanisms like unnatural splicing [34]. This is very similar situation to what we find in the Hox complex and may hint that transcriptional interference exists at other loci displaying a similar arrangement of genes. Examining the mechanism that mediates transcriptional interference at model loci like *iab-8* may help to define the conditions necessary for transcriptional interference to occur and potential lead to the identification other loci regulated in similar fashion.

## Materials and methods

Standard molecular biology techniques were used. Primers and G-blocks used for the construction of plasmids in this study are shown in S1 **and** S2 **Tables.**

### Fly strains

The *Canton S* strain was considered as *WT* for all studies. All strains were raised in standard cornmeal-agar food supplemented with propionic acid and nipagin. The crosses and egg laying for embryo collection were made at 25˚C, and the fly stocks were kept at 18˚C whenever possible.

Previously published mutations used in this work are: *elav^E476^* (kindly provided by C. Klämbt) [35], *Fab-8^64^* [36], *ΔmiR iab-8* [1], *del(ex3-8)* (also called *MSA deletion*, [19]), *inv(ex3-8)* (also called *MSA inversion*, [19]), *del(ex8)* [33] and *elav^cds20^*, *fne^Δ^* ([20, 37]).

### Construction of the *abdA*:*GFP* flies

The plasmid *pYex8abdAGFP* was first constructed by Gibson Assembly (Gibson, Young et al. 2009), using the "*Gibson Assembly Master Mix*" kit from NEB (Mass., USA). The backbone plasmid, *pY25*, a derivative of the *pw25* in which the *white* reporter was replaced by *yellow* [38]. The Gibson assembly required five fragments: 1. *pY25* linearized by *Not I* and *ClaI* digestion, 2. Gblock *attBexon8flk3'* containing an *attB* site and the region flanking the 3' end of the exon 8 of the *iab-8* ncRNA (see S1 Text **and** S2 Table), ordered from Eurofins MWG Operon, 3. A Kozak sequence and *EGFP* was amplified from the plasmid *pGSA-iab8* amplified with the primers *PYexC1GFPFw* and *PYex8C1ex8Fw* (see S1 Text **and** S1 Table). 4. Exon 8 of the *iab-8* ncRNA and initial 140bp of *abd-A*, was amplified from Drosophila genomic DNA with the primers *PYex8C1GFPRv* and *PYex8C1intFw* (see S1 Text **and** S1 Table), and 5. The region flanking the 5' end of the exon 8 of the *iab-8* ncRNA was amplified from Drosophila genomic DNA with the primers *PYex8C1ex8Rv* and *PYex8C1intRv* (see S1 Text **and** S1 Table).

This plasmid was then injected into embryos carrying the *del(ex8)* line (which contains an attB site in place of the *iab-8* exon 8) [33] and the *vas-PhiC31* integrase on the X [39].

## Is situ hybridization and immunostaining

*In situ* hybridizations and immunostainings were performed using standard protocols. The creation of all probes was done with the primers listed in **S3 Table** and the antibodies used in the immunostainings and in situs are listed in **S5 Table.**

## Microscopy

Fluorescence microscopy images were obtained on a Zeiss LSM710 or LSM800 confocal microscope and treated with the Fiji image analysis software [40] or a Zeiss Axioplan fluorescent microscope with an X-lite 120 lamp. All fluorescent colors presented are pseudo-colors generated by the Fiji software to the different channels collected. Confocal images are generally shown as maximum image projections from stacks of images, unless otherwise noted in the figure legends. Non-Fluorescent images were obtained using a Zeiss axiophot microscope using an Optronics camera and the MagnaFire v2.0 software. Images were cropped and mounted for figures in Microsoft Powerpoint and Adobe Photoshop.

## Annotation of the ELAV-binding coordinates in the *iab-8* ncRNA transcriptional unit

Chromosomal coordinates provided by [32] converted from the **dm3** coordinates to **dm6** coordinates (release of 2014) using the UCSC Batch Coordinate conversion tool (https://genome.ucsc.edu/cgi-bin/hgLiftOver). In the **dm6** release, the *iab-8* ncRNA coordinates correspond to: **chr3R:16831116–16923127**. ChIP-seq data for ELAV binding in coordinates comprised on this interval were inspected manually (For coordinates, see **S6** and **S7 Tables**).

## Analysis of differentially spliced transcripts

Total RNA from an overnight collection of embryos was used with primers *iab8 Ex6-7F* (spanning the iab-8 exon 6 to exon 7 junction) and *abdA Ex5R* (or *iab8 Ex8rev*) according to the instructions of the Qiagen One-Step RT-PCR kit. Products from the reaction (a mixture of bands, due to the differentially spliced products) were run on agarose gels and cloned into pGemTeasy (Promega Corp, Wisconsin, USA). Different sized inserts were selected and sequenced.

## mRNA-seq data processing and visualization

Published mRNA-seq data (GSE146986 [20]) were analyzed using the RNA-seq module from snakePipes [41] with default parameters. Reads were mapped to the dm6 genome annotation (Ensembl release 96) and visualized using IGV_2.8.2.

## Embryo collection and RT-qPCR

Embryos were raised and collected as described in Carrasco et al., 2020 [20]. Genotypes of embryos selected for RT-qPCR are: wild type: $w^{1118}/Y$. *Δelav*: $elav^{CDS20}/Y$. *ΔelavΔfne*: $elav^{CDS20}$, $fne^{Δ}/Y$. For each condition, 10–15 embryos were homogenized in TRIzol (Invitrogen) in 3 biological replicates, and total RNA was extracted according to the manufacturer's instructions. 300ng of total RNA were used for each RT-qPCR reaction. Reverse transcription used iScript gDNA Clear cDNA Synthesis Kit (Bio-Rad). RT-qPCR was performed in a LightCycler 480 II instrument using FastStart SYBR Green Master (Roche). RT-qPCR primer sequences are listed in **S1 Text** and **S4 Table**. Control reactions to measure the levels of *elav* and *fne* are shown in **S4 Fig**.

## Supporting information

**S1 Fig.** *in situ* hybridization using a probe binding to the *iab-8 ncRNA* exons 1 and 2 to examine transcription emanating from the *iab-8* ncRNA promoter in different genotypes. Collections of *wild-type* (WT) embryos, *del(ex3-8)* embryos and *inv(ex3-8)* embryos (genotypes marked on figure) are shown, stained with a probe to exons 1 and 2.
(TIFF)

**S2 Fig.** *in situ* hybridization using a probe located between the *iab-8 ncRNA* and the *abd-A* gene (*intergenic probe*) to examine read-through transcription from the *iab-8* ncRNA in PS 13 and 14 in different genotypes. **A.** Shows the genomic region with the location of the probe marked by a blue bar beneath the map. Stage 13/14 embryo from a *wild-type* (**B.**), *del (ex3-8)* (**C.**) and *inv(ex3-8)* (**D.**) embryos. Expression in the posterior CNS can be seen in *wild-type* (**B.**) and *del(ex3-8)* embryos (**C.**) but not in *inv(ex3-8)* (**D.**) embryos.
(TIFF)

**S3 Fig. Similar rates of transcription of the *abd-A* and *iab-8*ncRNA transcription units.** Similar rates of transcription of the *abd-A* and *iab-8* ncRNA transcription units are revealed in the CNS with the help of an intronic probe derived from intron 3 of the *abd-A* gene (indicated by the orange rectangle below the genomic map in A). B, note that while the patterns of expression detected from P7 to 12 is generated by the *abd-A* promoter, the pattern detected in PS13 and 14 originates from the *iab-8* ncRNA.
(TIFF)

**S4 Fig. *elav* and *fne* mRNA expression in embryos shown in Fig 7.** RT-qPCR quantification of *elav* and *fne* coding sequence (CDS) RNAs in wild-type, *Δelav*, and *Δelav Δfne* embryos. RNA was extracted from whole embryos 14-16h AEL (after egg laying) and 18-22h AEL. RNA levels were normalized to RpL32 (rp49) mRNA levels. Error bars represent mean ± SD of three biological replicates (10–15 embryos per replicate) for each genotype.
(TIFF)

**S1 Table. PCR primers for plasmid construction.**
(DOCX)

**S2 Table. GBlock used in the generation of the plasmid *pYex8abdAGFP*.**
(DOCX)

**S3 Table. Primers used to make *in situ* hybridization probes.**
(DOCX)

**S4 Table. Primers used for RT-qPCR.**
(DOCX)

**S5 Table. Primary antibodies used for immunostainings.**
(DOCX)

**S6 Table. Manual annotation of the genomic coordinates of the BX-C region between *abd-A* and *Abd-B* where ELAV binding has been detected by ChIP-seq in 6-8h embryos [32].**
(DOCX)

**S7 Table. Manual annotation of the genomic coordinates of the BX-C region between *abd-A* and *Abd-B* where ELAV binding has been detected by ChIP-seq in 10-12h embryos [32].**
(DOCX)

**S1 Text. Supplementary Materials and Methods.**
(DOCX)

## Acknowledgments

We thank Clément Immarigeon, Yohan Frei, Welcome Bender, Annick Mutero, and Emi Nagoshi for helpful discussion. We also thank Jorge Faustino, Eva Favre, Angel Glauser and the University of Geneva Bioimaging platform (Christophe Bauer and Jerome Bosset) for technical assistance. And finally, for particularly stimulating discussion and suggestions regarding the activity of ELAV, we thank Dr. Mathias Soller (University of Birmingham, UK). His input into this work was crucial and is sincerely appreciated.

## Author Contributions

**Conceptualization:** Javier J. Castro Alvarez, Daniel Pauli, François Karch, Robert K. Maeda.

**Data curation:** Javier J. Castro Alvarez, Judit Carrasco, Valérie Hilgers, Robert K. Maeda.

**Formal analysis:** Javier J. Castro Alvarez, Maxime Revel, Judit Carrasco, Fabienne Cléard, Daniel Pauli, François Karch, Robert K. Maeda.

**Funding acquisition:** Valérie Hilgers, François Karch, Robert K. Maeda.

**Investigation:** Javier J. Castro Alvarez, Maxime Revel, Judit Carrasco, Fabienne Cléard, Daniel Pauli, François Karch, Robert K. Maeda.

**Methodology:** Javier J. Castro Alvarez, Judit Carrasco.

**Project administration:** François Karch, Robert K. Maeda.

**Supervision:** Valérie Hilgers, François Karch, Robert K. Maeda.

**Validation:** Maxime Revel, Valérie Hilgers, François Karch, Robert K. Maeda.

**Visualization:** Judit Carrasco.

**Writing – original draft:** Javier J. Castro Alvarez, Robert K. Maeda.

**Writing – review & editing:** Javier J. Castro Alvarez, Maxime Revel, Judit Carrasco, Valérie Hilgers, François Karch, Robert K. Maeda.

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
