## [Decision Letter · Decision Letter 0]

6 Sep 2021

Dear Dr Maeda,

Thank you very much for submitting your Research Article entitled 'Repression of the Hox gene abd-A by ELAV-mediated Transcriptional Interference' to PLOS Genetics.

The manuscript was fully evaluated at the editorial level and by two independent peer reviewers. The reviewers appreciated the attention to an important topic but identified some concerns that we ask you address in a revised manuscript. 

In addition to the reviewer's comments we noticed that Panels B in Figures 1 & 4 are duplicated.  The images are both being used as WT references for their respective figures, so we assume the duplication was intentional.  We ask that you do one of two things.  Either, provide an independent WT control image for each figure (preferable), or note the duplication in the figure legend of Figure 4 (something like: "The wildtype panel in B is the same image displayed in figure 1B") so as to be transparent that the two images are the same.

We therefore ask you to modify the manuscript according to the review recommendations. Your revisions should address the specific points made by each reviewer.

[LINK]

Yours sincerely,

Gregory P. Copenhaver

Editor-in-Chief

PLOS Genetics

Reviewer's Responses to Questions

**Comments to the Authors:**

Reviewer #1: The MS by Castor Alvarez reports on the investigation of transcription mechanisms contributing to the exquisite precision of Hox gene expression patterns, known to shape the morphological architecture of bilaterian animals. The work presented is part of a large effort of this group to decipher the multiple mechanisms at play in the regulation of Drosophila Bithorax Hox gene expression.

This work precisely aims at unmasking the mechanisms that posteriorly restricts AbdA CNS expression to PS 12. Previous work had identified a long iab-8 ncRNA transcribed upstream of the AbdA gene. This iab-8 ncRNA includes a miRNA that provide only partial posterior repressive effect, suggesting additional mechanisms for AbdA CNS posterior repression. Here the authors explore how full repressive effect is achieved, as well as the basis for CNS specificity. The work that combines CRISPR genome editing with classical in situ exon-specific transcript expression analyses indicates that specifically in the posterior CNS (PS13), the iab-8 ncRNA extends into the AbdA transcription unit and produce transcriptional interference that shut down the activity of the AbdA promoter. The data supporting this view is strong and fully supports the author’s conclusion. The “synthetic” reproduction of the transcriptional interference” on the AbdA promoter EGFP reporter is very elegant and convincingly support this major finding. The work also suggests that CNS specificity in posterior repression is brought in by the CNS specific ELAV RNA binding protein. The data here also clearly support the conclusion, but given the somewhat weaker de-repression phenotype (see comment below), it may also suggest other involvements.

The relevance of the work is well introduced, experiments are well thought and properly documented in the MS, the rational beyond the experimental strategy is well explained and the overall data discussed within the broader frame of transcriptional interference and posterior dominance is also adequate.

In general, the MS elegantly uses the tools and extensive knowledge of Bithorax gene regulation to provide novel insights into the mechanisms at play in generating the exquisite precision of Hox gene expression.

In my opinion two points needs considerations

1-I feel the de-repression even in the context of elav, fne and mir-iab8 removal (Fig. 1G) does not phenocopy lack of repression induced by cis modifications (Fig. 1F). The authors disregard this difference that yet is apparent. This could be corrected either by considering it, or providing another illustration if this one is not best representing the data obtained. Also, it would be interesting to assess the impact of the elav, fne and mir-iab8 context using the intergenic probe and see if it results in a stronger loss of read through when compared to elav mutation alone (Fig. 6F).

2- The authors could discuss more explicitly the functional meaning that all except one read through transcript lack the initial AbdA ATG. If this read through transcript allows AbdA expression then how the whole mechanisms of transcriptional interference seems irrelevant to AbdA expression…Is this isoform very weakly expressed ?

Following are a few more minor suggestions the authors may take into account and that may help improving the MS.

1- The work strongly relies on previous work of the group to which the authors refers to. However stating what exactly in this previous work support the rational for the experimental strategy used in this MS would facilitate the reading. This is for example striking in the first section of the results part, where the authors indicate in two places that previous work suggested that the repressive element was located towards the 3’ end of the ncRNA (ref14). Summarizing the reason for that for the reader would be helpful.

2-Repression seems to occur in the late CNS. Why the RT PCR characterization of the iab-8 ncRNA extension into AbdA was performed starting from 0-24h embryos seems not properly justified. Would the diversity of transcripts characterized be lower if a later and shorter time window was selected?

3-In the second section of the result part, the authors indicate that they see a correlation between iab-8 ncRNA extension and abdA repression. It is not clear to me what support that claim ?

4-Also in the second section of the result part, the authors indicate that EGFP reporter construct “almost completely” re-creates the AbdA promoter region…”. The authors could be more specific indicating what is different …

5- Representing in Fig. 1 all the deletions generated and analyzed, and not only the largest one, could be of interest for the community.

6- The embryo displayed in panel D, Fig.S1 looks much older (all three gut constrictions are formed) than the others (gut not yet compartmentalized). To make sure differences are not due to that, an earlier embryo could be shown.

7- Figure 3: The authors could chose a representation where each transcript is displayed, so to more easily grasp the exact exon content of each of them.

Reviewer #2: It’s been long known that iab-8 ncRNA is the repressor of Hox gene abd-A in PS13 of the embryonic central nervous system. This mechanistic regulation is either achieved through the mir-iab-8 microRNA generated from iab-8, or postulated through a potential transcriptional interference. The manuscript by Alvarej Castro et al is aimed at providing experimental evidence for this proposed transcriptional inference. Using deletion lines and a gene reporter system (that nullifies mir-iab-8 impact), authors have shown abd-A repression in line with CNS specific 3’ extension of the iab-8 transcript. Authors have also shown that this extension (and repression) requires normal levels of Elav, a neuronal specific RNA binding protein. The experiments are very elegantly designed and executed with proper controls. The conclusions are well supported by the data. Accordingly, I have only minor comments

1. A recent study from the same group (Immarigeon et al., 2021) reported the expression of a secondary cell specific micropeptide from exon-8 of iab-8, with a reproductive function. Given the extension of iab-8 transcript beyond exon-8 in CNS, did authors check for such micropeptide-ORFs in the region of interest in the present study? Can such peptides mediate/participate in the repression and could those be targeted by Elav and/or fne?

2. It would be great if discussion can also dwell upon the functional relevance of such tissue specific repression mechanism in the context of iab-8/abd-A functions reported earlier.

3. Certain issues with figures/figure legends need to be fixed: For example, The figure legend for Figure 4 reads as “Each panel shows a Z-projection image made from confocal stacks of the posterior stage 13/14 CNS stained with: anti-ABD-A (in red in A., in grey in B.), anti-GFP (green C.-F.) and EN (blue in A. to show the anterior parasegment boundaries)”. However, Fig. 4A represents schematic while Fig. 4B (red) &C (grey) are the wild type. GFP is seen in Figs. 4D-F. Also EN is seen in Fig.4B but not in 4A. Abd-A-EGFP stained with EN could have been included. Also the relevance of using two wild type figures (red & grey) in this figure is not clear.

**Have all data underlying the figures and results presented in the manuscript been provided?**

Reviewer #1: Yes

Reviewer #2: Yes

PLOS authors have the option to publish the peer review history of their article (what does this mean?). If published, this will include your full peer review and any attached files.

Reviewer #1: No

Reviewer #2: No

---

## [Decision Letter · Decision Letter 1]

21 Oct 2021

Dear Dr Maeda,

We are pleased to inform you that your manuscript entitled "Repression of the Hox gene abd-A by ELAV-mediated Transcriptional Interference" has been editorially accepted for publication in PLOS Genetics. Congratulations!

Yours sincerely,

Gregory P. Copenhaver

Editor-in-Chief

PLOS Genetics

Comments from the reviewers (if applicable):

Reviewer's Responses to Questions

**Comments to the Authors:**

Reviewer #1: The authors have fully considered and discussed all requests.

Reviewer #2: Authors have addressed my concerns satisfactorily in the revised version.

**Have all data underlying the figures and results presented in the manuscript been provided?**

Reviewer #1: Yes

Reviewer #2: Yes

PLOS authors have the option to publish the peer review history of their article (what does this mean?). If published, this will include your full peer review and any attached files.

Reviewer #1: No

Reviewer #2: No

**Data Deposition**

http://datadryad.org/submit?journalID=pgenetics&manu=PGENETICS-D-21-00807R1

**Press Queries**

---

## [Editor Report · Acceptance letter]

9 Nov 2021

PGENETICS-D-21-00807R1 

Repression of the Hox gene abd-A by ELAV-mediated Transcriptional Interference 

Dear Dr Maeda, 

We are pleased to inform you that your manuscript entitled "Repression of the Hox gene abd-A by ELAV-mediated Transcriptional Interference" has been formally accepted for publication in PLOS Genetics! Your manuscript is now with our production department and you will be notified of the publication date in due course.

With kind regards,

Agnes Pap

PLOS Genetics

On behalf of:
